# RNA Interference-Mediated Suppression of Ecdysone Signaling Inhibits Choriogenesis in Two Coleoptera Species

**DOI:** 10.3390/ijms25084555

**Published:** 2024-04-22

**Authors:** Xiao-Qing Zhang, Lin Jin, Wen-Chao Guo, Kai-Yun Fu, Guo-Qing Li

**Affiliations:** 1Education Ministry Key Laboratory of Integrated Management of Crop Diseases and Pests, State & Local Joint Engineering Research Center of Green Pesticide Invention and Application, Department of Entomology, College of Plant Protection, Nanjing Agricultural University, Nanjing 210095, China; 2019202030@njau.edu.cn (X.-Q.Z.); jinlin@njau.edu.cn (L.J.); 2Key Laboratory of Intergraded Pest Management on Crops in Northwestern Oasis, Ministry of Agriculture, Urumqi 830091, China; gwc1966@163.com (W.-C.G.); fukaiyun000@foxmail.com (K.-Y.F.); 3Xinjiang Key Laboratory of Agricultural Biosafety, Institute of Plant Protection, Xinjiang Academy of Agricultural Sciences, Urumqi 830091, China

**Keywords:** *Leptinotarsa decemlineata*, *Henosepilachna vigintioctopunctata*, choriogenesis, ecdysone signal, chitin-like substance, eggshell

## Abstract

During choriogenesis in insects, chorion (eggshell) is formed by surrounding follicular epithelial cells in ovarioles. However, the regulatory endocrine factor(s) activating choriogenesis and the effect of chemical components on eggshell deserve further exploration. In two representative coleopterans, a coccinellid *Henosepilachna vigintioctopunctata* and a chrysomelid *Leptinotarsa decemlineata*, genes encoding the 20-hydroxyecdysone (20E) receptor heterodimer, ecdysone receptor (EcR) and ultraspiracle (USP), and two chitin biosynthesis enzymes UDP-N-acetylglucosamine pyrophosphorylase (UAP) and chitin synthase (ChS1), were highly expressed in ovaries of the young females. RNA interference (RNAi)-aided knockdown of either *HvEcR* or *Hvusp* in *H. vigintioctopunctata* inhibited oviposition, suppressed the expression of *HvChS1*, and lessened the positive signal of Calcofluor staining on the chorions, which suggests the reduction of a chitin-like substance (CLS) deposited on eggshells. Similarly, RNAi of *LdEcR* or *Ldusp* in *L. decemlineata* constrained oviposition, decreased the expression of *LdUAP1* and *LdChS1*, and reduced CLS contents in the resultant ovaries. Knockdown of *LdUAP1* or *LdChS1* caused similar defective phenotypes, i.e., reduced oviposition and CLS contents in the *L. decemlineata* ovaries. These results, for the first time, indicate that 20E signaling activates choriogenesis in two coleopteran species. Moreover, our findings suggest the deposition of a CLS on the chorions.

## 1. Introduction

The development of oocytes (oogenesis) in insect ovarioles consists of previtellogenesis, vitellogenesis and choriogenesis stages. Among the three successive physiological processes, choriogenesis is the final developmental period towards oocyte maturation, when the chorion (eggshell) is formed by the surrounding follicular epithelial cells [1]. Up to now, however, the regulatory endocrine factor(s) activating choriogenesis and the chemical components on eggshell have not been well explored.

Endocrine control of oogenesis has been widely reported in insects [2,3,4]. Nevertheless, all these publications are focused on the stimulation of previtellogenesis and vitellogenesis. Three classical hormones, ecdysteroids (20-hydroxyecdysone, 20E, the most active form), juvenile hormone (JH) and insulin-like peptides (ILP), have been confirmed as gonadotropins [2,5]. In *Tribolium castaneum*, for instance, application of JH III induces *vitellogenin* (*Vg*) mRNA in the previtellogenic females, whereas suppressing JH signal impairs *Vg* gene expression and Vg accumulation [5]. Similarly, RNAi studies show that 20E receptor heterodimer, ecdysone receptor and ultraspiracle genes (*EcR* and *usp*), and 20E downstream genes (*E75*, *E93*, *HR3*, *HR4* and *βFTZ-F1*) are required for vitellogenesis and oocyte maturation [6,7]. Moreover, bursicon neuropeptide (downstream 20E component) also triggers *Vg* expression, possibly through activation of JH and ILP pathways [8]. Up to now, it is unclear if the same endocrine pathways are involved in the regulation of choriogenesis. In fact, some researchers believe that insect choriogenesis is autonomously initiated when an oocyte has been developed to a certain vitellogenetic phase [9].

As for chemicals on insect eggshells, the most important component is proteins [1]. In the *Drosophila melanogaster* eggshells, for instance, more than 30 protein components are assembled [10,11]. Similarly, in the chorions of the superfamily Bombycoidea insects, more than 100 different polypeptides are accumulated in a stepwise fashion [1,12]. Intriguingly, a chitin-like substance (CLS) accumulates in oocytes during oogenesis, and it is also present in ovaries, along with newly laid and dark eggs in *Aedes aegypti* [13]. It needs to be established if the CLS is an eggshell structural component during oogenesis in insects other than *A. aegypti* [13].

Chitin biosynthesis has been well investigated in insects. UDP-N-acetylglucosamine pyrophosphorylase (UAP) and chitin synthases (ChS) catalyze the final two consecutive biochemical steps [14,15,16,17]. In most insects, ChSs have been segregated into two groups, namely ChS1 and ChS2. ChS1 is exclusively distributed in the ectodermally-derived epidermal cells, and it is responsible for chitin generation in epidermis, trachea, salivary gland, foregut and hindgut cells. ChS2 is specifically located in midgut cells and is dedicated to chitin formation in the peritrophic matrix (PM) [14,15,16,17]. In *Leptinotarsa decemlineata* larvae, UAPs are also encoded by *UAP1* and *UAP2* genes. UAP1 and UAP2 specifically catalyze chitin production in ectodermally-derived cells and midgut PM. Moreover, chitin synthesis is tightly controlled by 20E during larval development, through binding to EcR/USP heterodimer in *L. decemlineata* [17]. Providing that chitin biosynthesis is strictly coordinated within the cycle of insect molts [18,19], the biochemical process should also be governed by 20E cascade during metamorphosis in other insect species.

A chrysomelid *L. decemlineata* and a coccinellid *Henosepilachna vigintioctopunctata* are coleopteran potato defoliators. Our previous results reveal that vitellogenesis occurs in the 5- to 7-day-old females, while the choriogenesis takes place in the 8- and 9-day-old adults. Copulation and oviposition happen before and at the age of 8 days in *H. vigintioctopunctata* [20]. RNA interference (RNAi) of either *EcR* (using dsRNA derived from the common sequence of *EcRA* and *EcRB1*) or *usp* inhibits oocyte development, and dramatically represses the transcription of *Vg* in fat bodies and of *VgR* in ovaries in the two colcleopterans [21]. In the current paper, we performed the same RNAi experiments on the newly-emerged adult females. We intended to test two hypotheses. Firstly, 20E cascade mediates choriogenesis. Secondly, a CLS is deposited on oocyte chorion. Our findings supported the two assumptions.

## 2. Results

### 2.1. The Temporal Expression of Related Genes

Using quantitative reverse transcription polymerase chain reaction (qRT-PCR), the temporal expression patterns of two 20E receptor genes (*EcR* and *usp*) and three chitin biosynthesis genes (*UAP*, *ChS1* and *ChS2*) were respectively measured in the 0- to 9-, and 0- to 13-day-old ovaries in *L. decemlineata* and *H. vigintioctopunctata* (Figure 1). All 5 genes in *L. decemlineata* (*LdEcRA*, *LdEcRB1*, *Ldusp*, *LdUAP1*, *LdChS1*, *LdChS2*) and 4 genes in *H. vigintioctopunctata* (*HvEcRA*, *HvEcRB1*, *HvUAP*, *HvChS1*, *HvChS2*) were highly expressed in the young adults (at the ages of 0- to 5-days-old), whereas *Hvusp* was constantly transcribed in the 0- to 13-day-old *H. vigintioctopunctata* ovaries (Figure 1).

### 2.2. Knockdown of HvEcR Prevents Oviposition in H. vigintioctopunctata

Three days after ds*HvEcR* injection, the *HvEcR* level in the ovary was significantly reduced (Figure 2A). For the ds*egfp*-injected beetles, no female laid eggs until 8 days after adult emergence. They then continuously oviposited. In contrast, the dsHvEcR-injected females did not lay any eggs until 20 days post adult emergence. They then laid only turbid liquid (Figure 2B versus Figure 2C).

Consistent with failure of oviposition, the 30-day-old *HvEcR* RNAi females possessed larger body sizes and heavier fresh weights than the 30-day-old controls (Figure 2D–J). The mature eggs in the 30-day-old control ovaries were ellipse-shaped (Figure 2K). The average major axis and minor axis were 1.30 mm and 0.50 mm, respectively. The ratio of major/minor axis was 2.7 (Figure 2M–O). The eggs in the 30-day-old *HvEcR* RNAi ovaries were misshapen, nearly globular (Figure 2L); the mean major axis, minor axis and ratio of major/minor axis were 0.85 mm, 0.57 mm and 1.5, respectively (Figure 2M–O).

### 2.3. Depletion of Hvusp Affects Oogenesis in H. vigintioctopunctata

Injection of ds*Hvusp* into the 0-day-old females successfully knocked down the target gene in the ovary (Figure 3A). Whereas the control females (more than 8-days-old) continuously laid eggs (Figure 3B), the *Hvusp* RNAi females did not lay any eggs until around 20 days post eclosion. They then laid only turbid liquid (Figure 3C). Interestingly, some *Hvusp* RNAi females more than 40-days-old could lay a few normal eggs daily, along with turbid liquid (Figure 3D).

Similar to the 30-day-old *HvEcR* RNAi females, the 30-day-old *Hvusp* RNAi females had larger body sizes and heavier fresh weights than the 30-day-old controls (Figure 3E–G).

In the 20-day-old *Hvusp* RNAi ovarioles, there were two kinds of mature eggs. Many misshapen eggs were accumulated in the median oviduct and the genital chamber, whereas one or two normally-shaped eggs were sited in the lateral oviduct (Figure 3H).

The mature eggs in the 20-day-old control ovaries were ellipse-shaped (Figure 3I). The average major axis and minor axis were 1.25 mm and 0.45 mm, respectively. The ratio of major/minor axis was 2.8 (Figure 3K–M). Conversely, the eggs in the *Hvusp* RNAi ovarioles were misshapen (Figure 3J); the mean major axis, minor axis and ratio of major/minor axis were 1.30 mm, 0.70 mm and 1.8, respectively (Figure 3K–M).

### 2.4. RNAi of HvEcR Affected CLS Deposition on H. vigintioctopunctata Oocytes

Our previous results have documented the identification of two chitin synthase genes (*HvChS1* and *HvChS2*) in *H. vigintioctopunctata* [15]. Here we detected their expression levels in the 10- and 20-day-old ovaries. In contrast to *HvChS2*, which is only expressed in the midgut [15], the transcript levels of *HvChS1* were significantly reduced in the *HvEcR* RNAi samples, compared with those in the control ovaries (Figure 4A–D).

HE-stained sections of ovaries displayed that the maturing eggs in the control ovarioles had smooth and continuous chorions (Ch) (Figure 4E), whereas the eggshells in the *HvEcR* RNAi developing eggs were irregular and discontinuous (Figure 4F).

The presence of CLS was shown by Calcofluor staining [22]. The fluorescence images of frozen sections displayed that two layers of positive-stained membranes (↑) were observed in the control ovarioles, one lining the follicle cells (FC) and the other being deposited to the chorion (Ch) (Figure 4G,I). Conversely, only the positive-stained layer along the follicle cells was seen in the ds*HvEcR*-treated ovariole samples (Figure 4H,J).

### 2.5. Silence of Hvusp Blocked CLS Deposition on H. vigintioctopunctata Oocytes

CLS deposition on the *Hvusp* RNAi *H. vigintioctopunctata* oocytes was also compared (Figure 4K–R). The expression levels of *HvChS1* were significantly diminished in the 10- and 20-day-old Hvusp RNAi ovaries, compared with those in the control specimens (Figure 4K,L). In contrast, the transcript levels of HvChS2 varied little between *Hvusp* RNAi and control samples (Figure 4M,N).

From HE-stained sections, it was easily noted that the normal maturing eggs had smooth and continuous chorions (Ch) (Figure 4O). Conversely, the egg shells in the *Hvusp* RNAi eggs were irregular and discontinuous. Some yolk substances leaked out (^) (Figure 4P).

The fluorescence images of frozen sections stained with Calcofluor exhibited two layers of positively-stained membranes (↑) in the follicle cells (FC) and chorion (Ch) (Figure 4Q). Conversely, the two positively-stained layers almost disappeared in the dsHvusp-treated ovarian specimens (Figure 4R).

### 2.6. Knockdown of EcR/usp Prevents CLS Contents in L. decemlineata Ovaries

In the newly-eclosed *L. decemlineata* females, we repeated the RNAi experiments. The defective phenotypes were examined (Figure 5). Three days after ds*LdEcR* or ds*Ldusp* injection, the target genes *LdEcR* (Figure 5A) and *Ldusp* (Figure 5I) in the ovaries were respectively knocked down. When the 9-day-old control *L. decemlineata* females normally oviposited (Figure 5B), the 15-day-old *LdEcR* and *Ldusp* RNAi females laid turbid liquid, along with a few deformed eggs (Figure 5C). These misshapen eggs had shriveled eggshells and a wet surface (Figure 5C).

The expression of *LdUAP1*, *LdUAP2*, *LdChS1* and *LdChS2* in the 10-day-old ovaries was determined. The transcript levels of *LdUAP1* and *LdChS1*, but not *LdUAP2* and *LdChS2*, were significantly lowered in the *LdEcR* (Figure 5D–G) or *Ldusp* (Figure 5J–M) RNAi ovaries.

The chitin contents in the 10-day-old *LdEcR* (Figure 5H) or *Ldusp* (Figure 5N) RNAi ovaries were tested. As expected, the chitin amounts were greatly declined in the treated ovaries when evaluated by *N*-acetylglucosamine (GlcNAc) concentrations using the chitinase degraded method, compared with control specimens.

### 2.7. Chitin Defeciency Causes Similar Defects on L. decemlineata Oocytes

To determine if chitin shortage causes similar defects in oocytes to those in the *LdEcR* or *Ldusp* RNAi ovaries, we knocked down *LdUAP1*, *LdUAP2*, *LdChS1*, or *LdChS2* by injecting corresponding dsRNA into 4-day-old female adults (Figure 6A–D). Introduction of ds*LdUAP1*, ds*LdUAP2*, ds*LdChS1*, or ds*LdChS2* significantly lowered its target mRNA when detected 3 days post treatment, but displayed little influence on non-target transcripts (Figure 6A–D).

Depletion of *LdUAP1* and *LdChS1* significantly reduced chitin contents in the ovaries when measured with GlcNAc concentrations using the chitinase degraded method, in contrast to the control, and *LdUAP2* and *LdChS2* RNAi specimens (Figure 6E). Moreover, no mature oocytes were seen in 10-day-old *LdUAP1* and *LdChS1* RNAi ovaries, in contrast to the control, and *LdUAP2* and *LdChS2* RNAi samples (Figure 6G,I versus Figure 6F,H,J). A total of twenty days after treatment, the *LdUAP1* and *LdChS1* RNAi female adults began to lay turbid liquid, along with a few deformed eggs (Figure 6K).

## 3. Discussion

### 3.1. 20E Signal Triggers Choriogenesis

After initiation of choriogenesis, chorion proteins and other structural components are deposited on eggshells. However, the issue of if 20E signal activates choriogenesis is still an open question [9].

In the present paper, four pieces of experiment evidence were attained to indicate that 20E signal initiates choriogenesis in the two coleopterans. Firstly, both *EcR* and *usp* were actively expressed immediately after adult eclosion (Figure 1). The accumulated EcR and USP proteins may mediate 20E signaling and activate oogenesis. As a result, oocytes are continuously matured and the oviposition persists in mature *H. vigintioctopunctata* females [20]. Likewise, ecdysone response gene transcripts (*E75*, *E74*, *BrC*, *HR3*, *HR4*, and *FTZ-F1*) are mostly upregulated in the ovary during the first three days post blood meal in *Rhodnius prolixus* [23].

Secondly, our data showed that the *EcR* and *usp* transcripts were present in the ovaries in both *H. vigintioctopunctata* (Figure 1, Figure 2 and Figure 3) and *L. decemlineata* (Figure 1 and Figure 5) females, similar to those in *R. prolixus* [23,24].

Thirdly, knockdown of either *EcR* or *usp* significantly repressed oviposition in both *H. vigintioctopunctata* (Figure 2 and Figure 3) and *L. decemlineata* (Figure 5) females. In accordance with our data, knockdown of *EcR*, *usp*, *E75*, *E74* or *FTZ-F1* decreases mature eggs, and severely reduces the laid eggs and their hatching rate in *R. prolixus* [23,24]. The reduced oviposition in 20E signal repressed females has also been documented in *T. castaneum* [7], *D. melanogaster* [25], *Nilaparvata lugens* [26,27], and *Cimex lectularius* [28].

Lastly, we discovered that RNAi of either *EcR* or *usp* destroyed the structure of eggshells and deformed oocyte shapes (Figure 2, Figure 3 and Figure 5). Consistently, some of the eggs have an irregular shape and smaller volume laid by the *EcR*, *usp*, *E75*, *E74* or *FTZ-F1* RNAi *R. prolixus* females [23,24]. The deformation of chorion structure and oocyte shape gives a solid piece of experimental evidence that 20E cascade triggers choriogenesis, in addition to activating vitellogenesis in both *H. vigintioctopunctata* and *L. decemlineata* females [21].

### 3.2. 20E Signal Triggers the Production of a CLS

In the current article, our findings suggest that 20E signal triggers the production of a chitin-like substance (CLS), a component critical for the formation of oocyte chorion during choriogenesis in the two Coleoptera species.

Firstly, UAP and ChS catalyze the last two biochemical steps for chitin biosynthesis in *L. decemlineata* [16,17] and *H. vigintioctopunctata* [14,15]. In the present paper, expression analysis showed that their encoding genes were actively expressed in the adult ovaries (Figure 1, Figure 4, Figure 5 and Figure 6). Even though the expression levels of either UAP or ChS1 were high in the young females, a small rise occurred 5 to 7 days after emergence (Figure 1). The expression patterns are consistent with our previous result, that vitellogenesis initiates in the 5- to 7-day-old females of *H. vigintioctopunctata* [20]. Moreover, the expression of either *UAP* or *ChS* is strictly coordinated with the transcription of 20E receptor gene *EcR* in both coleopterans (Figure 1). Similar expression patterns between chitin biosynthesis and 20E cascade genes have been widely documented during larval development in representative insect species [17,22].

Secondly, biochemical analysis demonstrated that the CLS was present in *L. decemlineata* ovaries (Figure 5), similar to those on *A. aegypti* eggshells [13] and in *R. prolixus* ovaries [29]. Moreover, the Calcofluor stain [22] showed that two layers of CLS were observed in *H. vigintioctopunctata* ovaries, one lining the follicle cells and the other in the eggshells (Figure 4). Follicle cells are ectodermally-derived epidermal tissue. Thus, our data are in agreement with the common notion that ChS1 locates in the ectodermally-derived epidermal cells, while ChS2 is expressed in midgut cells [14,15,16,17].

Thirdly, our findings showed that knockdown of either *EcR* or *usp* significantly lowered the expression levels of *UAP* and *ChS1*, but not *ChS2*, in the two coleopterans (Figure 4 and Figure 5). Consistently, suppression of 20E signal reduced the CLS contents in *L. decemlineata* ovaries (Figure 5) and decreased the positive reaction to Calcofluor stain in *H. vigintioctopunctata* follicle cells and eggshells (Figure 4). Similarly, a CLS has been reported to be present in oocytes in a Diptera species [13]. Recently, a specific mucin protein has been documented to contribute to the formation of eggshell in *N. lugens* [30,31] and *Spodoptera exigua* [32]. Moreover, enzymes involved in chorion tanning are reported in *D. melanogaster* [33,34] and *N. lugens* [35] eggshells. Given that intestinal mucins exhibit a strong association with chitin [36] and tanning process involves the oxidative conjugation and cross-linking of cuticular proteins and chitin by quinones [37], it is reasonably assumed that many insect eggshells contains chitin. Further research will shed light on this issue.

Fourthly, knockdown of *LdUAP1* and *LdChS1*, rather than *LdUAP2* and *LdChS2*, caused similar defects in *L. decemlineata* oocytes to those in the *EcR* or *usp* RNAi beetles (Figure 6). We accordingly propose that dysfunction of EcR or USP inhibits CLS accumulation on the eggshells in *L. decemlineata*. As a result, the structure of eggshells is destroyed and the oocyte shapes are deformed (Figure 2, Figure 3 and Figure 5). Interestingly, ds*ChS* treatment affects oviposition in *R. prolixus* [29]. Knockdown of *ChS1* disrupts adult fecundity in *Aphis gossypii* [38]. Moreover, dietary introduction of chitin synthesis inhibitors declines fecundity in *A. aegypti* [13]. In addition, RNAi of *LsChS1* in pre-adult I results in deformation of ovaries and oocytes in adult *Lepeophtheirus salmonis* [39]. This result suggests the likelihood that CLS may also be present in eggshells and that deficiency of CLS causes misshapen eggs in other arthropods. Nevertheless, seeking to establish if 20E regulation of CLS affects deposition on oocyte chorion is widely acknowledged as a conservative way of engaging arthropods, and so this issue deserves further investigation in order to reach a final determination.

In the current paper, after we injected ds*LdUAP1* and ds*LdChS1* into 4-day-old female adults, the target mRNAs were significantly reduced three days after treatment (Figure 6). Our previous observation showed that some oocytes have completed vitellogenesis and that the patency among the follicle epithelium cells was closed in 5- to 7-day-old *H. vigintioctopunctata* females [20], similar to the oocyte maturation processes in other coleopterans [2]. Moreover, after vitellogenesis, oocytes are often arrested in prophase I of meiosis, showing very modest synthetic activity [40]. Therefore, it appears that only follicle epithelium cells can actively generate CLS in the 5- to 7-day-old *H. vigintioctopunctata* females and only the CLS biosynthesized by follicle epithelium cells can deposit on eggshells. Our findings support a common notion that the chorion is formed by the surrounding follicular epithelial cells during oocyte maturation [1].

### 3.3. Does 20E Signal Regulate the Production of Chorion Proteins?

Comparison of the data in the current survey displays two dissimilarities. Firstly, even though inhibition of chitin deposition affected eggshell structure in the two coleopterans after disruption of 20E signal, the defects of matured eggs are different. *L. decemlineata* eggs in treated females looked normal and could be deposited (Figure 5 and Figure 6), whereas *H. vigintioctopunctata* eggs were seriously deformed and failed to oviposit (Figure 2 and Figure 3). Secondly, within *H. vigintioctopunctata* the influence of chitin deposition was more severely impacted in the *usp* RNAi females (Figure 3), whereas the destruction of chorion and the delay of oviposition were more severe in the *EcR* RNAi females (Figure 2). The two dissimilarities suggest that 20E signal may also regulate the production of chorion proteins. In accordance with this suggestion, EcRB1 isoform and USP are responsible for 20E-mediated transcriptional activation of the eggshell gene *VM32E*, which produces a protein component of the vitelline membrane and endochorion layers [41]. Similarly, knockdown of *EcR*, *usp*, or ecdysone biosynthesis gene *shade* significantly reduces the expression of chorion protein gene transcripts (*Rp30* and *Rp45*) in *R. prolixus* [24].

If 20E signal stimulates the production of chorion proteins, as suggested above, the difference of eggshell phenotypic defects between *EcR* and *usp* RNAi females implies that EcR and USP respectively regulate a specific subset of chorion protein genes, dependent on, or independent of, EcR/USP complex (Figure 6L). In this context, both EcR and USP exert specific atypical function independent of EcR/USP complex. For EcR, it may dimerize with other partners, form homodimers [42] and/or form protein complexes that have yet to be described [43]. In *D. melanogaster* male adults, for example, EcR-depleted (but not USP-depleted) accessory glands fail to make seminal proteins and have dying cells. The active receptor may be a homodimer [44]. A similar paradigm is observed in the expression of genes encoding glue proteins in the salivary glands [43]. Interestingly, EcR apparently does not require USP as a heterodimeric partner in scorpions [45]. As for USP, it is able to form homomeric complexes in living cells [42]. It is also bound to other nuclear receptors [46]. In fact, some functions independent of EcR have been explored in *D. melanogaster* and *A. aegypti* USPs [46,47]. In *A. aegypti*, for instance, the interaction of Svp and USP, rather than binding competition for the Vg ecdysteroid response element, accounts for the inhibition of Vg expression after a batch of eggs is laid [48]. This issue deserves further investigation.

## 4. Materials and Methods

### 4.1. Insect Rearing

Both *H. vigintioctopunctata* and *L. decemlineata* beetles were cultured in laboratory at 28 ± 1 °C, 16 h:8 h light–dark photoperiod and 50–60% relative humidity conditions, using the fresh potato leaves, and using foliage at the vegetative growth or young tuber stages from potato plants (*Solanum tuberosum* L.) growing in insecticide-free soil. Under this feeding protocol, the duration from laying eggs to emergence adults of the next generation was about one month.

### 4.2. Synthesis of dsRNAs

A highly efficient and specific siRNA fragment originated from *EcR* or *usp* was selected using the siRNA online design website (http://sidirect2.rnai.jp/ (accessed on 15 June 2022)). Then, a pair of primers was designed using the software primer premier 5.0 (Appendix A) to amplify a cDNA including the siRNA fragment. The targeted cDNA sequence was further BLASTN searched against transcriptome data to identify any possible off-target sequences that had an identical match of 20 bp or more. Moreover, a cDNA fragment was derived from enhanced green fluorescent protein (*egfp*) in *Aequorea victoria*. These fragments were respectively amplified by PCR using specific primers (Appendix A) conjugated with the T7 RNA polymerase promoter. The dsRNAs were synthesized using the MEGAscript T7 High Yield Transcription Kit (Ambion, Austin, TX, USA) according to the manufacturer’s instructions. Subsequently, the synthesized dsRNA (at a concentration of 5–8 μg/μL) was determined by agarose gel electrophoresis and the NanoDrop 2000 spectrophotometer (Thermo Fisher Scientific, New York, NY, USA) and kept at −80 °C until use.

### 4.3. Injection of dsRNAs

The same method described previously was used to inject dsRNA into 0-day-old *H. vigintioctopunctata* and *L. decemlineata* female adults [14]. Briefly, an aliquot (0.1 μL) of the solution, including 400 ng dsRNA, was injected into the newly emerged female adults. Negative control newly emerged female adults were injected with the same volume of dsegfp solution.

For 0-day-old *H. vigintioctopunctata* female adults, two biologically independent experiments were carried out using different generations, with each bioassay setting two treatments: (1) ds*egfp* and (2) ds*HvEcR* or ds*Hvusp*. A group of 10 injected newly emerged female adults was set as a replicate. The resultant newly emerged female adults were allowed to feed potato foliage until death. Each treatment had 24 replicates. A total of 9 replicates were harvested 3, 10 and 20 days after treatment to dissect their ovaries. Three replicates were sampled 3 days after injection for qRT-PCR, to test RNAi efficacy in the ovaries. The other 6 replicates were used to measure the expression levels of genes involved in chitin biosynthesis (*HvChS1* and *HvChS2*) in the ovaries. Three replicates were used to record oviposition, measure body size and detect fresh weight. Another 12 replicates were collected, dissected for collection of the fresh oocytes in the ovaries, and for observation under a microscope or stained using HE or Calcofluor 10, 20, 30 or 40 days after injection.

For 0-day-old *L. decemlineata* female adults, two biologically independent experiments were carried out using different generations, with two treatments: (1) ds*egfp* and (2) ds*LdEcR* or ds*Ldusp*. A group of 10 injected newly emerged female adults was set as a replicate. The resultant newly emerged female adults were allowed to feed potato foliage until death. Each treatment had 12 replicates. A total of 6 replicates were harvested 3 and 10 days after treatment to dissect their ovaries. Three replicates were sampled 3 days after injection for qRT-PCR to test RNAi efficacy in the ovaries. Three replicates at the age of 10 days old were used to measure the expression levels of genes involved in chitin biosynthesis (*LdUAP1*, *LdUAP2*, *LdChS1* and *LdChS2*) in the ovaries. Three replicates were used to record oviposition. Another three replicates were collected 10 days after injection, and the ovaries were dissected to determine chitin content.

For 4-day-old *L. decemlineata* female adults, a bioassay was performed with five treatments: (1) ds*egfp*, (2) ds*LdUAP1*, (3) ds*LdUAP2*, (4) ds*LdChS1* and (3) ds*LdChS2*. A group of 10 injected female adults was set as a replicate, and were allowed to feed potato foliage. Each treatment had 12 replicates. Three replicates were harvested 3 days after treatment for qRT-PCR to test RNAi efficacy, and for measurement of the expression levels of genes involved in chitin biosynthesis (*LdUAP1*, *LdUAP2*, *LdChS1* and *LdChS2*) in the ovaries. Three replicates were used to record oviposition. Another six replicates were collected 10 days after injection, and the ovaries were then dissected and chitin content in the ovaries was determined.

### 4.4. Quantitative Reverse Transcription PCR (qRT-PCR)

For temporal expression analysis, RNA templates were derived from 0–9- and 0–13-day-old *L. decemlineata* and *H. vigintioctopunctata* female adults, respectively. Each sample contained 20–30 individuals and was repeated three times. For analysis of the effects of treatments, total RNA was extracted from treated female adults. Each sample contained 10 individuals and was repeated three times. The RNA was extracted using SV Total RNA Isolation System Kit (Promega, Madison, WI, USA). Purified RNA was subjected to DNase I to remove any residual DNA according to the manufacturer’s instructions. Quantitative mRNA measurements were performed by qRT-PCR in technical triplicate, using 2 stably-expressed internal control genes (for *L. decemlineata*, *LdRP18* and *LdRP4*; for *H. vigintioctopunctata*, *HvRPS18* and *HvRPL13*; the primers listed in Appendix A) according to the published results [49,50]. An RT negative control (without reverse transcriptase) and a non-template negative control were included for each primer set to confirm the absence of genomic DNA and to check for primer-dimer or contamination in the reactions, respectively. The primer pair for each gene was tested with a 5-fold logarithmic dilution of a cDNA mixture to generate a linear standard curve (crossing point plotted vs. log of template concentration), which was used to calculate the primer pair efficiency.

All primer pairs amplified a single PCR product with the expected sizes, showed a slope less than −3.0 and exhibited efficiency values ranging from 2.4 to 2.7. Data were analyzed by the 2^−ΔΔCT^ method, using the geometric mean of the four internal control genes for normalization.

### 4.5. Hematoxylin-Eosin (HE) Staining

HE staining was performed to observe defective phenotypes in *H. vigintioctopunctata* ovaries. Briefly, the ovaries in the ds*egfp*- and ds*Hvusp*-treated larvae were dissected 10, 20 and 30 days after the initiation of bioassay, and were then fixed in 4% paraformaldehyde and embedded in paraffin. The two embedded tissues were subsequently cut into 6-μm thick sections. The sections were stained using Mayer’s H&E (Yeasen, Shanghai, China) following a routine staining procedure and observed with an Olympus BH-2 light microscope (Olympus, Tokyo, Japan).

### 4.6. Chitin Staining with Calcofluor

The *H. vigintioctopunctata* ovary sections prepared above were also stained with Calcofluor according to a previously described method [22]. Briefly, section preparations were incubated in 0.001% Calcofluor in 100 mm Tris-HCl, pH 9 for 30 min at room temperature. After washing off the excess Calcofluor, the fluorescence was recorded using light of appropriate excitation and emission wavelengths.

### 4.7. Chitin Analysis

The same method as described in [16,17] was used to test chitin contents in 10-day-old *L. decemlineata* ovaries. Briefly, the samples were individually mixed with 0.5 g zirconium beads (0.7 mm diameter, BioSpec Products, Bartlesville, OK, USA) and 0.5 mL 6% KOH, and were homogenized. The homogenates were then heated at 80 °C for 90 min, and were centrifuged at 12,000× *g* for 20 min and the supernatants were removed. The pellets were individually suspended in 1 mL PBS, and centrifuged again at 12,000× *g* for 20 min and the PBSs were discarded. Each pellet was then resuspended in 200 µL Mcllvaine’s buffer (0.1 M citric acid, 0.2 M NaH_2_PO_4_, pH 6) and 5 µL of Streptomyces plicatus chitinase-63 (5 mg/mL in PBS) was added to hydrolyze chitin to GlcNAc by incubation for 40 h at 37 °C.

GlcNAc concentrations were individually measured by a modified Morgan–Elson assay [51]. In a 0.2 mL PCR tube, 10 µL 0.27 M sodium borate and 10 µL of sample supernatant (12,000× *g*, 1 min centrifugation) were combined. In a thermocycler, samples were heated to 99.9 °C for about 60 s, mixed gently, and incubated at 99.9 °C for 10 min. Immediately upon cooling to room temperature, 100 µL of diluted dimethylaminobenzaldehyde (DMAB) solution (10% *w*/*v* DMAB in 12.5 mL concentrated HCl and 87.5 mL glacial acetic acid stock, diluted 1: 10 with glacial acetic acid) was added, followed by incubation at 37 °C for 20 min. A total of 80 µL of each sample was transferred to 96-well low-evaporation microtitre dishes, and the absorbance at 585 nm was recorded. Standard curves were prepared from stocks of 0.075–2.0 mM GlcNAc.

### 4.8. Data Analysis

IBM SPSS Statistics 27 (https://www.ibm.com/products/spss-statistics accessed on 23 July 2022) was used for statistical analyses. The averages (±SE) were compared using *t*-test, or analysis of variance with the Tukey-Kramer test.

## 5. Conclusions

Accordingly, a presumptive model is proposed for summarization of the regulation of 20E signal to choriongenesis in two representative Coleoptera species (Figure 6L). Our findings highlighted that 20E cascade triggers the deposition of chorion chemical components on the developing oocytes. Moreover, we uncovered that a CLS is critical for oocyte maturation. Our research provides a solid foundation for future studies on hormone regulation of reproduction in beetles.

## Figures and Tables

**Figure 1 ijms-25-04555-f001:**
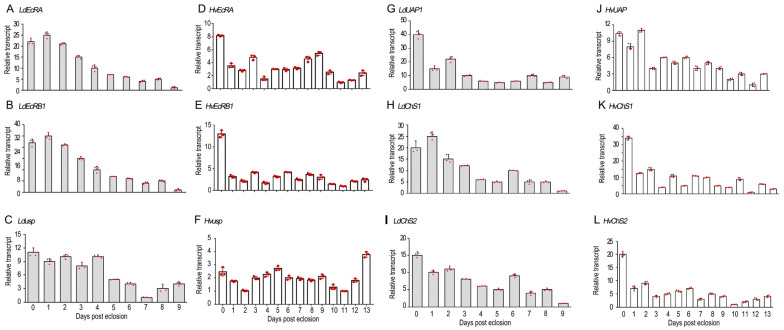
The temporal expression patterns of selected genes in the ovaries of two Coleoptera potato defoliators. The cDNA templates were derived from the ovaries of 0- to 9-day-old *L. decemlineata*, and 0- to 13-day-old *H. vigintioctopunctata* female adults, respectively. The transcript levels of *EcRA*, *EcRB1*, *usp*, *UAP*, *ChS1*, and *ChS2* were determined. For each sample, 3 independent pools of 5–10 individuals (red dots) were measured in technical triplicate using real-time quantitative PCR. The bars represent 2^−ΔΔCT^ value (±SD) normalized to the geometrical mean of house-keeping gene expression. The lowest transcript levels at a specific developmental time point were set as 1.

**Figure 2 ijms-25-04555-f002:**
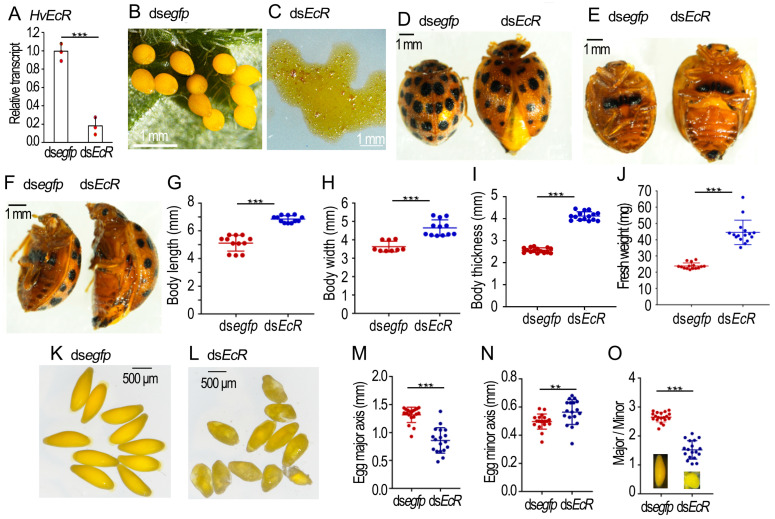
Knockdown of *HvEcR* in the 0-day-old *Henosepilachna vigintioctopunctata* female adults. The 0-day-old female adults were injected with 0.1 μL ds*egfp* or ds*HvEcR* (400 ng). The treated beetles were fed on fresh potato foliage. (**A**) Expression of *HvEcR* three days after treatment (n = 3, red dots). Relative transcript is the ratio of relative copy number in treated individuals to ds*egfp*-treated controls, which is set as 1. (**G**–**J**) Comparison of body sizes and fresh weights in the 30-day-old females (n = 9–12). (**M**–**O**) The egg sizes in the 30-day-old females (n = 15). The averages and SD ranges are drawn. Different stars indicate significant difference at *p* value < 0.01 (**), or 0.001 (***) using *t* test. (**B**–**F**,**K**,**L**) showed oviposited eggs, body sizes and egg shapes from 30-day-old females.

**Figure 3 ijms-25-04555-f003:**
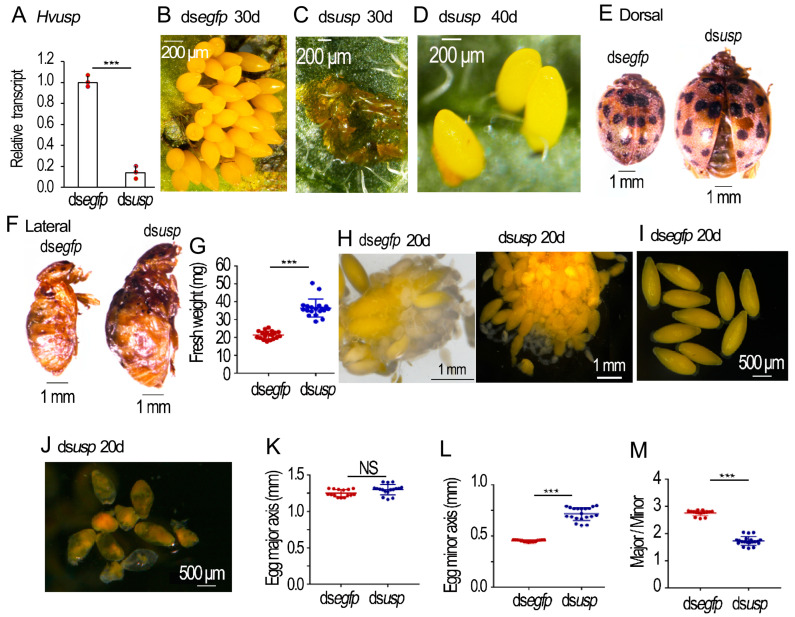
Silence of *Hvusp* in the 0-day-old *Henosepilachna vigintioctopunctata* female adults. The 0-day-old female adults were injected with 0.1 μL ds*egfp* and ds*Hvusp* (400 ng). The treated beetles were fed on fresh potato foliage. (**A**) Expression of *Hvusp* three days after treatment (n = 3, red dots). Relative transcript is the ratio of relative copy number in treated individuals to dsegfp-treated controls, which is set as 1. (**B**–**D**), (**E**,**F**) and (**H**–**J**) respectively showed oviposited eggs by 30- or 40-day-old females, body sizes of 30-day-old females and egg shapes from 20-day-old females. (**G**) The fresh weights of the 30-day-old females (n = 18). (**K**–**M**) The egg sizes of the 20-day-old females (n = 15). The averages and SD ranges are drawn. Different stars indicate significant difference at *p* value < 0.001 (***) using *t* test. NS, no significance.

**Figure 4 ijms-25-04555-f004:**
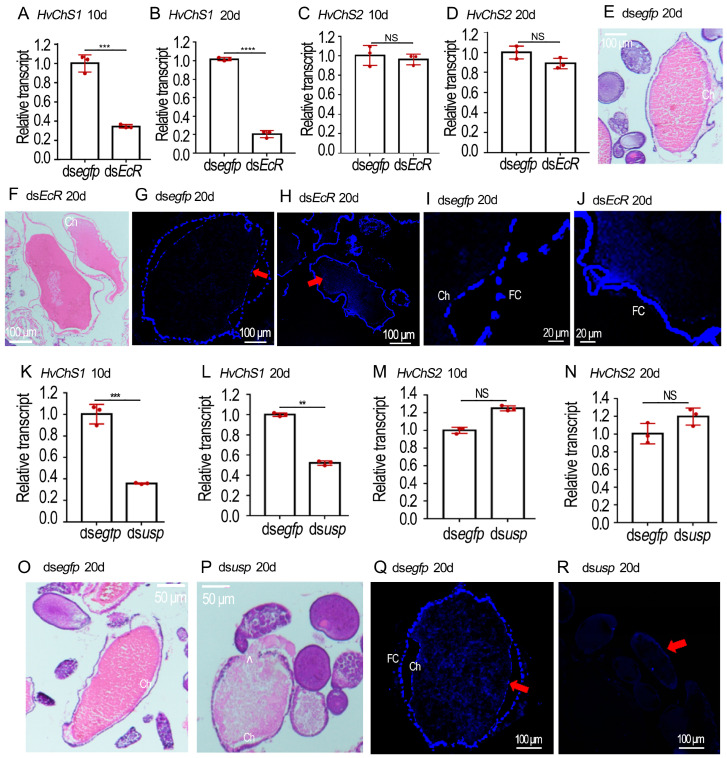
RNAi of *HvEcR* or *Hvusp* represses choriogenesis in *Henosepilachna vigintioctopunctata*. (**A**–**D**,**K**–**N**) Expression of HvChS1 and HvChS2 from ovaries 10 and 20 days post injection of ds*egfp* and ds*HvEcR*, or ds*egfp* and ds*Hvusp* using the 2^−ΔΔCT^ method (n = 3, red dots). Relative transcripts are the ratios of relative copy numbers in treated individuals to ds*egfp* controls, which are set as 1. The columns represent averages, with vertical lines indicating SD. Different stars indicate significant difference at *p* value < 0.01 (**), 0.001 (***) or 0.0001 (****) using *t* test. NS, no significance. (**E**,**F**,**O**,**P**) The paraffin sections of ovaries from 20-day-old female adults, treated with hematoxylin-eosin staining. (**G**–**J**,**Q**,**R**) The paraffin sections of ovaries from 20-day-old female adults, dyed with Calcofluor-White and 10% KOH. Blue color in (**G**–**J**,**Q**,**R**) (shown with red arrow) marks chitin layer. FC, the follicle cells; Ch, chorions.

**Figure 5 ijms-25-04555-f005:**
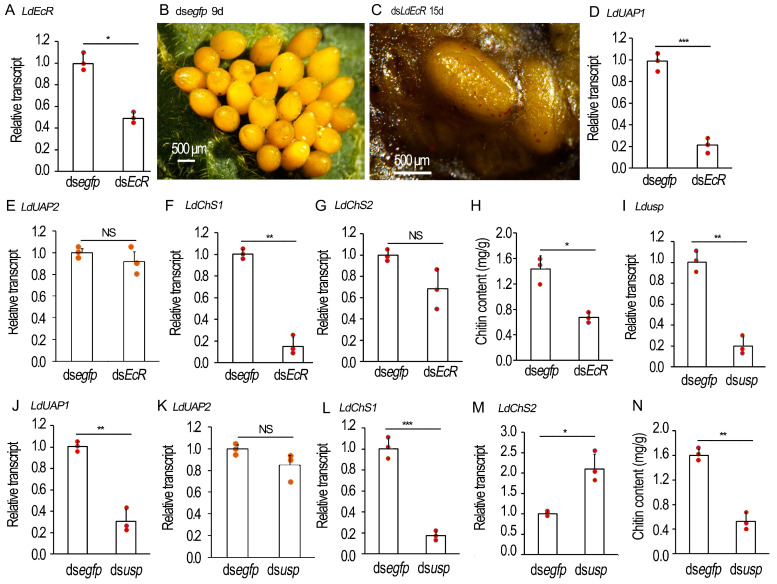
RNAi of *LdEcR* or *Ldusp* in the 0-day-old *Leptinotarsa decemlineata* female adults. Two biologically independent experiments were carried out using different generations, with two treatments: (1) ds*egfp* and (2) ds*LdEcR* or ds*Ldusp*. The treated beetles were fed on fresh potato foliage. (**A**,**D**–**G**,**I**–**M**) Expression of *LdEcR*, *LdUAP1*, *LdUAP2*, *LdChS1* and *LdChS2*, or *Ldusp*, *LdUAP1*, *LdUAP2*, *LdChS1* and *LdChS2* in the ovaries three or ten days after treatment (n = 3, red dots). Relative transcripts are the ratios of relative copy numbers in treated individuals to ds*egfp*-treated controls, which are set as 1. The chitin contents of the 10-day-old ovaries were measured by *N*-acetylglucosamine (GlcNAc) concentrations using the chitinase degraded method (**H**,**N**). The columns represent averages, with vertical lines indicating SD. Different stars indicate significant difference at *p* value < 0.05 (*), 0.01 (**), or 0.001 (***) using *t* test. NS, no significance. (**B**,**C**) The oviposited eggs by 15-day-old adult females.

**Figure 6 ijms-25-04555-f006:**
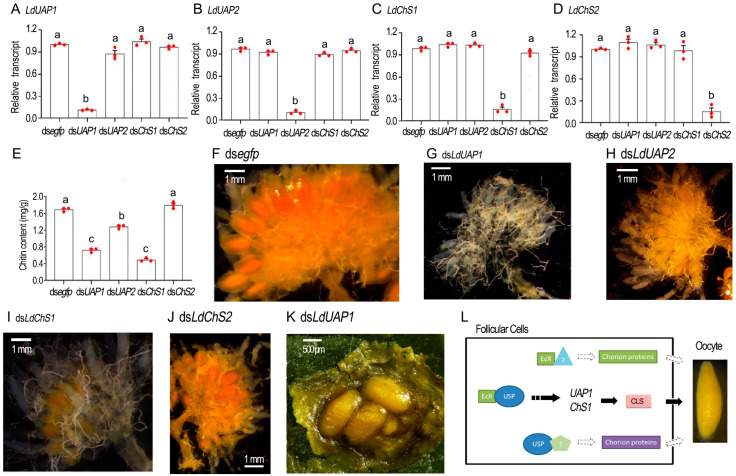
Knockdown of chitin biosynthesis genes in the 4-day-old *Leptinotarsa decemlineata* female adults. The 4-day-old female adults were injected with 0.1 μL ds*egfp*, ds*LdUAP1*, ds*LdUAP2*, ds*LdChS1* or ds*LdChS2* (400 ng). The treated beetles were fed on fresh potato foliage. (**A**–**D**) Expression of *LdUAP1*, *LdUAP2*, *LdChS1* and *LdChS2* three days after treatment (n = 3, red dots). Relative transcripts are the ratios of relative copy numbers in treated individuals to ds*egfp*-treated controls, which are set as 1. (**E**) The chitin contents of the 10-day-old ovaries measured by *N*-acetylglucosamine (GlcNAc) concentrations using the chitinase degraded method. The columns represent averages, with vertical lines indicating SD. Different letters in (**A**–**E**) panels indicate significant difference at *p* value < 0.05 using analysis of variance with the Tukey-Kramer test. (**F**–**J**) The 10-day-old ovaries. (**K**) The oviposited eggs by 20-day-old females. (**L**) A presumptive model of the molecular regulation of 20E signal for choriogenesis in the two coleopteran species. EcR/USP complex activates the expression of chitin biosynthesis genes, such as UAP1 and ChS1 in *L. decemlineata* or ChS1 in *H. vigintioctopunctata*, or may indirectly act on these genes, to regulate the supply of chitin-like substance (CLS). Meanwhile, EcR and USP may respectively regulate a specific subset of chorion protein genes, dependent or independent of EcR/USP complex, to synthesize constructive proteins for the formation of eggshells.

## Data Availability

Data generated in association with this study are available in the Appendix A published online with this article.

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
