# Peer review of "RNA Interference-Mediated Suppression of Ecdysone Signaling Inhibits Choriogenesis in Two Coleoptera Species"

_ijms, 2024, doi:10.3390/ijms25084555_

Round 1
Reviewer 1 Report
Comments and Suggestions for Authors
Presented manuscript describes novel and interesting data regarding involvement of ecdysone signaling in choriogenesis and depositing a chitin-like structure (CLS) to oocytes in two Coleopterans, a Henosepilachna vigintioctopunctata and a Leptinotarsa decemlineata. Experiments demonstrated that knockdown of EcR and Usp nuclear receptors, key proteins for the ecdysone signaling indeed impaired fecundity of both species and decreased the transcriptional level of enzymes involved in chitin synthesis. Microscopy revealed impairment of CLS of oocytes upon knockdowns. I believe the experimental work is solid and the manuscript will be of interest to readers because it clarifies the under-described role of ecdysone signaling in adult physiology.
I have three major and a few minor suggestions to the authors:
Major:
1) In the Figure 6 H and N you presented the data of EcRi and Uspi influence on the chitin content in the eggs – that is a direct confirmation of knockdown influence on CLS deposition in L. decemlineata. Can you (if possible) provide the data of chitin content for knockdowns of H. vigintioctopunctata. I understand that it may be more difficult as in that specie knockdowns had more severe influence on the fecundity, but as you provided some analysis data for the eggs after knockdown (like in Figure 2 M-O), so it may be done.
2) In the scheme of your model (Figure 7L) and in the discussion you mentioned that you showed the influence of the EcRi and Uspi on UAP transcription in both species. While I failed to find data regarding UAP for H. vigintioctopunctata. Please show it or remove the statement (in the second case it may be changed to influence on UAP1 in L. decemlineata).
3) Please add a section in the discussion which help to correlate the temporal profiles from Figure1 describing the transcription of genes after eclosion with their role in chorion deposition. If I understand, in the species you studied the active chorion deposition start a few days after eclosion (close to the date of oviposition) – this is supported with your recent article where you showed that choriogenesis starts 8-9 days after eclosion in H. vigintioctopunctata (https://www.sciencedirect.com/science/article/pii/S1226861523001474?via%3Dihub)
While your current data show that transcription peak of EcR, Usp and even ChS and UAP happens 1 day after eclosion. So, it is not correlated with active choriogenesis. This is may be OK about EcR and Usp, as these proteins may be accumulated before but influence later upon the increase in 20E concentration (which profile in fact would be extremely interesting to know). But the facts of ChS and UAP transcription in 1 day after eclosion and their involvement in choriogenesis do not add up. Could you please provide some ideas how ChS and UAP may be stored and acted later (may be in some inactive state)?
Minor:
1) Please group the graphs in some way - according to genes or species
2) Please indicate how you collected the eggs you analyzed in Figure 2L.
3) Figure 4 and further – regarding HvChS2. Is it possible to assess the absolute level of transcription of this gene—whether it has changed because it was not transcribed in the ovaries or whether it was transcribed at a high level but did not change? (this question also applies to other sections – to LdChS2 and LdUAP2
4) Why were chitinase transcription levels assessed at 10 days and 20 days, although the transcription profile showed greater transcription in the first days after hatching? Why not measure the level of gene activity on days 2-3 after eclosion, when RNAi is already known to work, but the endogenous level of genes itself is still expected to be high (experiments on L. decemlineata confirm that 3 days is a sufficient period for RNAi to work). Please explain in the text the choice of points 10 and 20 days
Author Response
Major:
1) In the Figure 6 H and N you presented the data of EcRi and Uspi influence on the chitin content in the eggs – that is a direct confirmation of knockdown influence on CLS deposition in L. decemlineata. Can you (if possible) provide the data of chitin content for knockdowns of H. vigintioctopunctata. I understand that it may be more difficult as in that specie knockdowns had more severe influence on the fecundity, but as you provided some analysis data for the eggs after knockdown (like in Figure 2 M-O), so it may be done.
Response: Since the same scientific method may cause similar systemic error even if it is used in different insect species, the best way to avoid the systemic error is using different method in various species to get the same results. Therefore, we have used different methods to display the influence of EcRi or Uspi on CLS deposition in different species in the manuscript. We do not think it is necessary to repeat the measurement of chitin content in H. vigintioctopunctata, as that in L. decemlineata.
2) In the scheme of your model (Figure 7L) and in the discussion you mentioned that you showed the influence of the EcRi and Uspi on UAP transcription in both species. While I failed to find data regarding UAP for H. vigintioctopunctata. Please show it or remove the statement (in the second case it may be changed to influence on UAP1 in L. decemlineata).
Response: It is true. We have added some words in the legend to clearly describe the results obtained in the manuscript. Moreover, we have changed Figure 7L in the new version.
3) Please add a section in the discussion which help to correlate the temporal profiles from Figure1 describing the transcription of genes after eclosion with their role in chorion deposition. If I understand, in the species you studied the active chorion deposition start a few days after eclosion (close to the date of oviposition) – this is supported with your recent article where you showed that choriogenesis starts 8-9 days after eclosion in H. vigintioctopunctata (https://www.sciencedirect.com/science/article/pii/S1226861523001474?via%3Dihub)
Response: A good advice. We have added several sentences to compare our expression data with the results in our recent publication. As follows:
Even though the expression levels of either UAP or ChS1 were high at the young females, a small rise occurred 5 to 7 days after emergence (Figure 1). The expression patterns are consistent with our previous result, that vitellogenesis initiates in the 5- to 7-day-old females of H. vigintioctopunctata [20].
While your current data show that transcription peak of EcR, Usp and even ChS and UAP happens 1 day after eclosion. So, it is not correlated with active choriogenesis. This is may be OK about EcR and Usp, as these proteins may be accumulated before but influence later upon the increase in 20E concentration (which profile in fact would be extremely interesting to know). But the facts of ChS and UAP transcription in 1 day after eclosion and their involvement in choriogenesis do not add up. Could you please provide some ideas how ChS and UAP may be stored and acted later (may be in some inactive state)?
Response: Firstly, from our response to Major concern 3, a small rise of UAP or ChS1 occurred 5 to 7 days after emergence is consistent with our previous result, that vitellogenesis initiates in the 5- to 7-day-old females of H. vigintioctopunctata [20]. Secondly, in our previous results, the highest level of either HvChS1 or HvUSP in whole body is measured in the 0-day-old adult, the content is dramatically reduced in the 1-day-old adult in H. vigintioctopunctata [14, 15], similar to the result in the ovary in this manuscript. However, the adults may spend at least 4 days to finish chitin deposition and tanning of the integument due to the adults do not ingest potato foliage within 4 days post emergence (indicating by decreased fresh weight) [20]. Therefore, both HvChS1 and HvUSP mRNAs may normally function or ChS and UAP proteins may accumulate for several days to exert their roles in CLS biosynthesis during choriogenesis 5 to 7 days after emergence.
Minor:
1) Please group the graphs in some way - according to genes or species
Response: Another good advice. We have grouped Figure 4 and 5 together and 6 figures were remained in the new version.
2) Please indicate how you collected the eggs you analyzed in Figure 2L.
Response: We have added several words to describe the egg collection in the new version.
3) Figure 4 and further – regarding HvChS2. Is it possible to assess the absolute level of transcription of this gene—whether it has changed because it was not transcribed in the ovaries or whether it was transcribed at a high level but did not change? (this question also applies to other sections – to LdChS2 and LdUAP2
Response: According to the expression patterns of ChS2 in Figure 1, the highest levels were 15 to 20 times more than the lowest. It appears that ChS2 is actively transcribed in the ovaries.
4) Why were chitinase transcription levels assessed at 10 days and 20 days, although the transcription profile showed greater transcription in the first days after hatching? Why not measure the level of gene activity on days 2-3 after eclosion, when RNAi is already known to work, but the endogenous level of genes itself is still expected to be high (experiments on L. decemlineata confirm that 3 days is a sufficient period for RNAi to work). Please explain in the text the choice of points 10 and 20 days
Response: It appears a misunderstanding. In fact, the 4-day-old female adults were injected with 0.1 μL dsegfp, dsLdUAP1, dsLdUAP2, dsLdChS1 or dsLdChS2 (400 ng). The treated beetles were fed on fresh potato foliage. Three days after treatment, transcript levels of LdUAP1, LdUAP2, LdChS1 and LdChS2 were determined. The chitin contents of the 10-day-old ovaries were measured by N-acetylglucosamine (GlcNAc) concentrations using the chitinase degraded method. The oviposition was observed 0 to 20 days post eclosion. See figure legend in Figure 6.
Reviewer 2 Report
Comments and Suggestions for Authors
In this study, the authors examined the roles of ecdysone receptors and Usp in regulating choriogenesis. Previous studies had demonstrated that knockdown of these genes inhibit vitellogenesis and the lack of eggs laying could be reflective of this. The beetles however eventually start laying eggs and it is in these eggs that they observed decreased expression of chitin synthase 1 and UDP-N-acetylglucosamine pyrophosphorylase. These results along with some observations using microscopy led to the conclusion that 20E is needed for the accumulation of chitin-like substance on chorions. Overall, the study seems to be done well, and the work is clearly and logically presented. I have a few suggestions that should be relatively easy to address.
Major comments
- There are isoforms of EcRs that the authors examine. This should be clearly discussed in the introduction.
- In the qPCR data, are the red dots for each bar distinct biological replicates? They all seem extremely similar for biological replicates. Please explain what the red dots represent in the legend.
- The figure legends could be reorganized to facilitate reading. It’s best to state what each panel is showing right after the title rather than at the very end. I would also start the sentence with the panel label: e.g. (A-C) Expression of EcRA , EcRB and usp in L. decemlineata. (D-F)… etc
- The panels in Fig. 1 are also ordered a little oddly with A, B, C… going down. Usually the panels go side ways. It’s not a big issue but a bit unusual.
- Figure 3: It might be good to show the dsegfp-injected ovaries for comparison?
- Figure 7: Are all of the images taken at the same magnification? I think this should be stated or a scale should be provided.
- Figure 7L: I find this model a little confusing. What is the basis by which the authors think there should be the unknown factor represented by the triangle and the pentagon? And why should EcR and Usp be acting separately? Since they both seem to regulate choriogenesis, wouldn’t’ the simplest interpretation be to have EcR and Usp heterodimerizing to regulate chorion proteins?
Minor comments
- Some grammatical issues exist. I’ve listed what I could find below but it’s not an exhaustive list. Please reread the manuscript carefully.
- There are several paragraphs with just one sentence. There should be at least three in each paragraph. Please go through the document and fix those.
- Line 17 “Coccinellid” and “Chrysomelid” should be lower case.
- Line 25: “declined” – suggest change to “reduced”
- Line 28, 303: “Coleopteran” should be lower case
- Line 28: “Moreover, our findings propose the deposition of a CLS on the chorions.” reads a bit weird. Maybe “suggest” instead of “propose”?
- Line 51: what is meant by “government of Choriogenesis”?
- Line 66: Is there a reference for this?
- Line 108: Italicize “dsHvEcR”
- Line 127; 1555: Instead of “were shown”, I would suggest “are shown”.
- Line 180 & 205: delate “The images were showed.”
- Fig. 4 legend: Define FC and Ch.
- Line 182: “Silence of” – replace with Knockdown of since it is not a complete removal of expression
- Line 189: “Some yolk substances were seeped out (^)”sounds a bit odd. Maybe change to “Some yolk substances leaked out”?
- Line 207: “ Shortage of 20E signal” I don’t think this is the best description. I would suggest “knockdown of EcR”
- Line 211: Change “silenced” to “knocked down”
- Line 214: “shrinked eggshells”? What do you mean by this?
- Line 313: “while ChS2 in midgut cells” – maybe change to “while ChS2 is expressed in midgut cells”
- Line 325: “researches” should be singular
- Line 351: what do you mean by “tune”?
- Line 356: what do you mean by “failed to lay out”?
- Line 357: “the influence of chitin deposition was more serious” Odd phrasing Do you mean “the influence of chitin deposition was more severely impacted”?
- Line 363: “silence of” “knockdown of”?
Comments on the Quality of English Language
There are some grammatical issues that need be fixed and words that are confusing. I listed some above but please reread the work and make necessary changes so it doesn't detract from an otherwise well-executed study.
Author Response
Major comments
- There are isoforms of EcRs that the authors examine. This should be clearly discussed in the introduction.
Response: We have added several words in the Introduction to clearly describe the isoforms in the new version. As follows:
RNA interference (RNAi) of either EcR (using dsRNA derived from the common sequence of EcRA and EcRB1) or usp inhibits oocyte development, dramatically represses the transcription of Vg in fat bodies and of VgR in ovaries in the two Coleopterans [21].
- In the qPCR data, are the red dots for each bar distinct biological replicates? They all seem extremely similar for biological replicates. Please explain what the red dots represent in the legend.
Response: Yes. The red dots for each bar indicate biological replicates. We have added the explanation about the red dots from Figure 1 through Figure 6 in the new version.
- The figure legends could be reorganized to facilitate reading. It’s best to state what each panel is showing right after the title rather than at the very end. I would also start the sentence with the panel label: e.g. (A-C) Expression of EcRA , EcRB and usp in L. decemlineata. (D-F)… etc
Response: A good advice. We have rewritten all the figure legends accordingly in the new version.
- The panels in Fig. 1 are also ordered a little oddly with A, B, C… going down. Usually the panels go side ways. It’s not a big issue but a bit unusual.
Response: We prefer to keep the order.
- Figure 3: It might be good to show the dsegfp-injected ovaries for comparison?
Response: We have added a photo from the dsegfp-injected ovaries in the new version.
- Figure 7: Are all of the images taken at the same magnification? I think this should be stated or a scale should be provided.
Response: We have provided the scales in Figure 7 in the new version.
- Figure 7L: I find this model a little confusing. What is the basis by which the authors think there should be the unknown factor represented by the triangle and the pentagon? And why should EcR and Usp be acting separately? Since they both seem to regulate choriogenesis, wouldn’t’ the simplest interpretation be to have EcR and Usp heterodimerizing to regulate chorion proteins?
Response: We have explained the model in the last two paragraphs of the Discussion. Briefly, comparison of the data in the current survey displays two dissimilarities. Firstly, even though inhibition of chitin deposition affected eggshell structure in the two Cole-opterans after disruption of 20E signal, the defects of matured eggs are different. L. de-cemlineata eggs in treated females looked normal and could be deposited (Figure 65, 76), whereas H. vigintioctopunctata eggs were seriously deformed and failed to lay out (Figure 2, 3). Secondly, within H. vigintioctopunctata the influence of chitin deposition was more serious in the usp RNAi females (Figure 3), whereas the destruction of chorion and the delay of oviposition were more severe in the EcR RNAi females (Figure 2). The two dis-similarities suggest that 20E signal may also tune the production of chorion proteins through a pathway independent of EcR/USP complex.
In this context, both EcR and USP exert specific atypical function independent of EcR/USP complex. For EcR, it may dimerize with other partners, form homodimers [42] and/or form protein complexes that have yet to be described [43]. In D. melanogaster male adults, for example, EcR-depleted (but not USP-depleted) accessory glands fail to make seminal proteins and have dying cells. The active receptor may be a homodimer [44]. A similar paradigm is observed in the expression of genes encoding glue proteins in the salivary glands [43]. Interestingly, EcR apparently does not require USP as a heterodimeric partner in scorpions [45]. As for USP, it is able to form homomeric complexes in living cells [42]. It is also bound to other nuclear receptors [46]. In fact, some functions are independent of EcR have been explored in D. melanogaster and A. aegypti USPs [46, 47]. In A. aegypti, for in-stance, the interaction of Svp and USP rather than binding competition for the Vg ec-dysteroid response element accounts for the inhibition of Vg expression after a batch of eggs is laid [48]. This issue deserves further investigations.
Minor comments
- Some grammatical issues exist. I’ve listed what I could find below but it’s not an exhaustive list. Please reread the manuscript carefully.
Response: Thanks a lot.
- There are several paragraphs with just one sentence. There should be at least three in each paragraph. Please go through the document and fix those.
Response: We have checked throughout the manuscript and rewritten the sentences.
- Line 17 “Coccinellid” and “Chrysomelid” should be lower case.
Response: We have changed accordingly in the new version.
- Line 25: “declined” – suggest change to “reduced”
Response: We have changed accordingly in the new version.
- Line 28, 303: “Coleopteran” should be lower case
Response: We have changed accordingly in the new version.
- Line 28: “Moreover, our findings propose the deposition of a CLS on the chorions.” reads a bit weird. Maybe “suggest” instead of “propose”?
Response: We have changed accordingly in the new version.
- Line 51: what is meant by “government of Choriogenesis”?
Response: We have replaced ‘government’ with ‘regulation’ in the new version.
- Line 66: Is there a reference for this?
Response: We have listed all references in the following sentences within this paragraph.
- Line 108: Italicize “dsHvEcR”
Response: We have changed accordingly in the new version.
- Line 127; 1555: Instead of “were shown”, I would suggest “are shown”.
Response: According to the advice in the major concern, we have rewritten all the figure legends in the new version.
- Line 180 & 205: delate “The images were showed.”
Response: According to the advice in the major concern, we have rewritten all the figure legends in the new version.
- Fig. 4 legend: Define FC and Ch.
Response: We have added the definition in the new version.
- Line 182: “Silence of” – replace with Knockdown of since it is not a complete removal of expression
Response: We have replaced ‘silence’ with ‘knockdown’ or ‘RNAi’ throughout the manuscript.
- Line 189: “Some yolk substances were seeped out (^)”sounds a bit odd. Maybe change to “Some yolk substances leaked out”?
Response: We have changed accordingly in the new version.
- Line 207: “ Shortage of 20E signal” I don’t think this is the best description. I would suggest “knockdown of EcR”
Response: We have changed accordingly in the new version.
- Line 211: Change “silenced” to “knocked down”
Response: We have replaced ‘silence’ with ‘knockdown’ or ‘RNAi’ throughout the manuscript.
- Line 214: “shrinked eggshells”? What do you mean by this?
Response: We have replaced ‘shrinked’ with ‘shriveled’ in the manuscript.
- Line 313: “while ChS2 in midgut cells” – maybe change to “while ChS2 is expressed in midgut cells”
Response: We have changed accordingly in the new version.
- Line 325: “researches” should be singular
Response: We have changed accordingly in the new version.
- Line 351: what do you mean by “tune”?
Response: We have replaced ‘tune’ with ‘regulate’ in the manuscript.
- Line 356: what do you mean by “failed to lay out”?
Response: We have replaced ‘failed to lay out’ with ‘failed to oviposit’ in the manuscript.
- Line 357: “the influence of chitin deposition was more serious” Odd phrasing Do you mean “the influence of chitin deposition was more severely impacted”?
Response: We have changed accordingly in the new version.
- Line 363: “silence of” “knockdown of”?
Response: We have replaced ‘silence’ with ‘knockdown’ or ‘RNAi’ throughout the manuscript.
Round 2
Reviewer 1 Report
Comments and Suggestions for Authors
The authors took into account all the comments. I recommend accepting the article in its current form